# Cyclic Adenosine Monophosphate in Cardiac and Sympathoadrenal GLP-1 Receptor Signaling: Focus on Anti-Inflammatory Effects

**DOI:** 10.3390/pharmaceutics16060693

**Published:** 2024-05-22

**Authors:** Anastasios Lymperopoulos, Jordana I. Borges, Renee A. Stoicovy

**Affiliations:** Laboratory for the Study of Neurohormonal Control of the Circulation, Department of Pharmaceutical Sciences, Barry and Judy Silverman College of Pharmacy, Nova Southeastern University, Fort Lauderdale, FL 33328-2018, USA; jb3837@mynsu.nova.edu (J.I.B.); rs2981@mynsu.nova.edu (R.A.S.)

**Keywords:** adrenal chromaffin cell, cardiac, central nervous system, cyclic AMP, Epac, GLP1 receptor, inflammation, protein kinase A, signal transduction, sympathetic nervous system

## Abstract

Glucagon-like peptide-1 (GLP-1) is a multifunctional incretin hormone with various physiological effects beyond its well-characterized effect of stimulating glucose-dependent insulin secretion in the pancreas. An emerging role for GLP-1 and its receptor, GLP-1R, in brain neuroprotection and in the suppression of inflammation, has been documented in recent years. GLP-1R is a G protein-coupled receptor (GPCR) that couples to Gs proteins that stimulate the production of the second messenger cyclic 3’,5’-adenosine monophosphate (cAMP). cAMP, acting through its two main effectors, protein kinase A (PKA) and exchange protein directly activated by cAMP (Epac), exerts several anti-inflammatory (and some pro-inflammatory) effects in cells, depending on the cell type. The present review discusses the cAMP-dependent molecular signaling pathways elicited by the GLP-1R in cardiomyocytes, cardiac fibroblasts, central neurons, and even in adrenal chromaffin cells, with a particular focus on those that lead to anti-inflammatory effects by the GLP-1R. Fully elucidating the role cAMP plays in GLP-1R’s anti-inflammatory properties can lead to new and more precise targets for drug development and/or provide the foundation for novel therapeutic combinations of the GLP-1R agonist medications currently on the market with other classes of drugs for additive anti-inflammatory effect.

## 1. Introduction

Glucagon-like peptide-1 (GLP-1) is one of the most important incretins produced in the body, stimulating post-prandial insulin secretion from the pancreas and participating in the physiological regulation of blood glucose levels and glucose availability to peripheral tissues [1]. Glucose-dependent insulinotropic polypeptide, formerly known as gastric inhibitory polypeptide (GIP), is another important endogenous incretin [2]. GLP-1 is a 37 amino acid-long polypeptide synthesized and secreted from the L cells in the brush border of the small and the large intestines [3]. GLP-1 is synthesized as a fragment of the much larger pre-hormone polypeptide proglucagon, which contains glucagon, GLP-2, and other incretin hormone fragments [3]. GLP-1 is continuously cleaved from proglucagon and secreted from the intestinal L cells in small amounts. However, meal ingestion markedly boosts its secretion. Its action is terminated via degradation by the enzyme dipeptidyl-peptidase 4 (DPP-4) in the portal vein and liver [1]. In reality, less than 15% of secreted GLP-1 enters systemic circulation. Nevertheless, GLP-1 is now known to act also as a neurotransmitter in the central nervous system (CNS), specifically in brainstem neurons projecting to hypothalamic areas that regulate metabolic homeostasis and energy balance (food intake and energy consumption) [4].

GLP-1 acts through a G protein-coupled receptor (GPCR) termed the GLP-1 receptor (GLP-1R) [5,6]. GLP-1R is a member of the class B of GPCRs and, more specifically, of the B1 (secretin receptors) subfamily (the other being the B2 subfamily that comprises adhesion receptors) [7]. Thus, GLP-1R is a single polypeptide chain with seven hydrophobic transmembrane alpha-helices, crucial for G protein interaction and activation [8,9], a rather long extracellular N-terminus forming the main binding region to the agonist (GLP-1), an intracellular C-terminus, and three extracellular (ECLs) and three intracellular loops (ICLs) [6,7,8,9]. Unlike the class A GPCRs, the vast majority of which have a short N-terminus and large C-termini or ICL3s, the human GLP-1R has a very long N-terminus (141 amino acids), and short ICLs (8, 13, and 23 amino acids for ILC1, ICL2, and ICL3, respectively) [6]. It also has a relatively long ECL1 (31 amino acids) that also participates in hormone binding [6]. The endogenous hormone GLP-1 first makes contacts with the N-terminal region, away from the cell membrane-embedded heptahelical core of the receptor, but, subsequently, engages all three ECLs of the receptor to activate it [6,8]. Upon agonist activation, GLP-1R interacts with the alpha subunit of the stimulatory Gs type of heterotrimeric G proteins, which then exchanges guanosine diphosphate (GDP) for guanosine triphosphate (GTP) to become activated [10]. Upon this guanine nucleotide exchange, G_αs_ dissociates from the G_βγ_ dimer and the receptor, and instead interacts with and activates the cell membrane-residing enzyme adenylyl cyclase (AC), which converts adenosine triphosphate (ATP) into the second messenger cyclic 3’,5’-adenosine monophosphate (cAMP) [10].

The GLP-1R is expressed in pancreatic β-cells where it mediates GLP-1-induced, cAMP-mediated insulin secretion but, importantly, it is now known to be present at physiologically significant levels in a variety of extra-pancreatic tissues and cell types, such as pulmonary epithelial cells, cardiac myocytes, gastric and intestinal mucosa, and, as mentioned above, in CNS neurons of various regions in the brain, including the autonomic nervous system. Indeed, vagus nerve (cholinergic nervous system) ganglia branching into the nucleus tractus solitarius (NTS) of the brainstem, together with signals from the hypothalamus, thalamus, and the hemispheres, play a crucial role in the regulation of all essential organ functions (cardiovascular, respiratory, etc.) [1,11,12]. The effects of GLP-1R on metabolic regulation and energy balance, such as stimulation of insulin and inhibition of glucagon secretions, anti-apoptotic protection of pancreatic β-cells, CNS-mediated regulation of food intake, appetite, and energy consumption, modulation of gastric emptying/digestive function, and body weight reduction, among others, are well documented [1,2,3,4]. In addition, GLP-1R exerts a variety of beneficial cardiovascular actions, such as lipoprotein lowering and protection against atherosclerosis, anti-hypertensive effects, and reduction in cardiovascular inflammation [13,14].

Importantly, the amount of evidence for a beneficial protective effect of GLP-1 and its analogs currently used in diabetes and obesity therapies (the “so-called” GLP-1R agonists, GLP1-RAs) against both systemic and tissue/organ-specific inflammations has been surging in recent years. Indeed, GLP1-RAs have been documented to reduce inflammation in the cardiovascular system, in the brain, in the gastrointestinal tract, in the airways, in white adipose tissue cells, and in other organs/tissues (reviewed in [3,14,15,16,17,18,19,20]). Yet, the molecular signaling mechanisms that underlie these anti-inflammatory effects of the GLP-1R are still under intense investigation and continue to be uncovered. Given that the GLP-1R appears to signal exclusively through the Gs protein/AC/cAMP signaling axis, cAMP must play a central role in the anti-inflammatory signaling of this receptor. The present review provides an account of the current literature pertaining to cAMP-dependent anti-inflammatory signaling of the GLP-1R, highlighting important knowledge gaps and emerging targets for anti-inflammatory therapy along the way. 

## 2. Chemistry of GLP-1R Agonists

As expected, the GLP-1Rs’ affinity for GLP-1 is extremely high (in the nM range) [6]. In contrast, other proglucagon-derived or incretin-related peptide hormones [glucagon, GIP, pituitary adenylate cyclase-activating polypeptide (PACAP), vasoactive intestinal polypeptide (VIP)], have minimal or no affinity for GLP-1R at all [6]. Exendin-4 [1,2,3,4,5,6,7,8,9,10,11,12,13,14,15,16,17,18,19,20,21,22,23,24,25,26,27,28,29,30,31,32,33,34,35,36,37,38,39], a peptide originally found in the venom of the Arizona desert lizard, *Heloderma suspectum*, shares ~53% of structural homology with GLP-1 and is about equipotent at GLP-1R activation [6]. Interestingly, an exendin-4 fragment lacking the first eight amino acids of exendin-4, exendin [9,10,11,12,13,14,15,16,17,18,19,20,21,22,23,24,25,26,27,28,29,30,31,32,33,34,35,36,37,38,39], acts as a full antagonist of the GLP-1R [6]. Exendin-4 [1,2,3,4,5,6,7,8,9,10,11,12,13,14,15,16,17,18,19,20,21,22,23,24,25,26,27,28,29,30,31,32,33,34,35,36,37,38,39] mimics almost all physiological actions of GLP-1 but, contrary to GLP-1, it is resistant to enzymatic degradation by the endogenous metabolic enzyme of GLP-1, dipeptidyl peptidase (DPP)-4 [6,21]. Due to the DPP4 action, GLP-1’s half-life is very short [21]. To combat this problem, different semi-synthetic GLP-1 analogs have been developed with longer half-lives and improved bio-availabilities (Figure 1). These GLP-1R agonist drugs (GLP1-RAs) are broadly categorized into two subgroups: 

(a) Agents that are protected from DPP4 degradation. These include analogs of exendin-4 (the “natides”), e.g., exenatide (Figure 1), and lixisenatide [22]. The 39 amino acid-long exenatide was the first synthetic GLP-1 RA to be FDA-approved [21]. Exendin-4 analogs are DPP4-resistant because they lack the Ala^2^ (and have Gly instead) necessary for DPP-4-mediated cleavage (Figure 1); in other words, GLP-1 is a DPP-4 substrate (Figure 1) but exendin-4 is not. This requirement for DPP-4 degradation, however, led to the development of DPP-4-resistant human GLP-1 analogs, as well as that of the “glutides”: semaglutide, dulaglutide, and albiglutide (Figure 1) [23]. Albiglutide and dulaglutide have simply the Ala^2^ replaced with Gly^2^, while semaglutide (the “blockbuster” drug “Ozempic”) contains a 2-amino-isobutyric acid “spacer” between His^1^ and Ala^2^, which, as an unnatural amino acid, blocks DPP-4 action on the molecule (Figure 1).

(b) Agents that exhibit enhanced stability and bioavailability by means of slowing down renal elimination [23]. Liraglutide is the prototypic GLP1-RA of this subgroup, having the 16-carbon atom palmitic acid chain esterified to the Lys^20^ via a glutamic acid linker (Figure 1). Liraglutide served as a lead for semaglutide’s development, since semaglutide also contains a (much larger) fatty di-acid chain (containing the 18-carbon atom stearic acid) attached to Lys^20^ via a glutamic acid linker (Figure 1). These fatty acid modifications of liraglutide and semaglutide serve to increase binding affinity to serum albumin, which effectively slows down renal elimination. Of note, human GLP-1 contains two lysines (Lys^20^ and Lys^34^), so, in order to ensure that the fatty acid chain is attached to Lys^20^ only, Lys^34^ was replaced with arginine (Arg^34^) in both liraglutide and semaglutide (Figure 1). Therefore, the only major difference between liraglutide and semaglutide is that the former is still a DPP-4 substrate, while the latter is not, and thus, semaglutide has an even longer half life (~7 days) than liraglutide does (~13 h) [23] (Figure 1). In other words, liraglutide is only renal elimination-resistant and belongs to the second subgroup, whereas semaglutide is both DPP4- and renal elimination-resistant (belongs to both subgroups). Similarly to semaglutide, dulaglutide and albiglutide are both DPP4- and renal elimination-resistant but, unlike semaglutide, their delayed renal elimination is due to their conjugation with human albumin (for albiglutide) or due to the dimerization of two identical GLP-1 analog polypeptide chains (~90% homologous to human GLP-1), each conjugated with an immunoglobulin IgG class Fc fragment (crystallizable fragment) through which the dimerization occurs (for dulaglutide) (Figure 1) [23]. These conjugations increase the size of the molecule dramatically, thereby slowing down its renal excretion. Notably, there is also an orally available formulation of semaglutide (Rybelsus) on the market, in which semaglutide is co-formulated with a gastrointestinal absorption enhancer, sodium *N*-(8-[2-hydroxybenzoyl]-amino)-caprylate (SNAC) [24]. SNAC prevents enzymatic degradation and facilitates the absorption of the GLP1-RA by transiently opening interepithelial tight junctions to facilitate intercellular transport, which helps the peptide minimize its exposure to the proteolytic degradation-causing highly acidic environment of the stomach lumen, thereby increasing absorption through the gastric mucosa [24]. SNAC also slightly solubilizes fatty membranes, further facilitating semaglutide transport and absorption. Once absorbed into the systemic circulation, SNAC dissociates from semaglutide, “freeing” the GLP1-RA to interact with receptors [24].

Finally, the latest generation of GLP-RAs consists of agents that activate other receptors in addition to the GLP-1R, the so-called “multi-receptor agonists” [21]. The additional receptors they activate respond to other incretin hormones structurally related to GLP-1. Tirzepatide is the prototypic agent of this “multi-agonist” class, activating both the GLP-1R and the GIP receptor (GLP-1R/GIPR dual agonist) (Figure 1) [21]. Tirzepatide is based on exendin-4, its C-terminus is amidated, and is also DPP4-resistant, containing two separate 2-amino-isobytyric acid spacers and a very large fatty acid chain attached to Lys^20^ for delayed renal elimination (Figure 1). Retatrutide is another “multi-agonist”, structurally similar to tirzepatide, which activates the glucagon receptor in addition to the GLP-1R and the GIPR [25].

## 3. cAMP in Inflammation

As mentioned above, cAMP is the second messenger produced inside the cell in response to GLP-1R activation and mediates almost all of the intracellular signaling elicited by this receptor. cAMP plays essential roles in the regulation of almost every aspect of cellular homeostasis in every mammalian cell type, including immune cell migration, activation, proliferation, and survival [26]. Therefore, cAMP is a master regulator of inflammatory processes [26,27]. cAMP has several effectors in cells, including cyclic nucleotide-gated (CNG) ion channels and popeye domain-containing (POPDC) proteins [10,28,29]. However, the versatile Ser/Thr protein kinase A (PKA), also known as cAMP-dependent protein kinase, and the exchange protein directly activated by cAMP (Epac)-1/2, originally discovered as guanine nucleotide exchange factor (GEF) for the monomeric G protein (small guanosine triphosphatase, GTPase) Ras-associated protein (Rap)-1, are its main effectors in most cells [10,28] (Figure 2). cAMP-activated PKA regulates (activates or inhibits) numerous other proteins/enzymes via phosphorylation, with the transcription factor CREB (cAMP response element (CRE)-binding protein) being one of its most prominent substrates (Figure 2) [26]. Dependent on the cell type, GLP-1R-stimulated cAMP elevation has been reported to result also in raised free intracellular calcium concentration, regulation of voltage-dependent and inwardly rectifying (hyperpolarizing) potassium channels, and activation of several kinases, in addition to PKA, such as extracellular signal-regulated kinase (ERK)1/2 and other mitogen-activated protein kinases (MAPKs), protein kinase C (PKC), phosphoinositide 3’-kinase (PI3K), and protein kinase B (PKB or Akt) [6].

cAMP levels inside the cell at any given time are a function of the balance between two activities: that of AC that synthesizes cAMP, and of that of the phosphodiesterases (PDEs) that degrade cAMP, terminating its action, i.e., PDE1, -2, -3, -4, -7, -8, -10, or -11 [10]. PDE4 in particular, a cAMP-specific PDE, i.e., a PDE that only degrades cAMP and not cyclic guanosine monophosphate (cGMP), is elevated in several inflammatory cells and conditions, and thus tightly regulates cAMP levels in inflammatory states (Figure 2) [26,30]. Indeed, at the onset of inflammation, PDE4 is upregulated, opposing cAMP elevations inside the immune cell [30]. CREB activated via PKA-mediated phosphorylation translocates from the cytoplasm to the nucleus, promoting the expression of inflammation pro-resolving mediators, anti-inflammatory cytokines (such as interleukin (IL)-10), and macrophage polarization (mediated by IL-10), efferocytosis, and granulocyte apoptosis, all of which result in inflammation resolution (Figure 2) [26]. At the same time, PKA inhibits the transcriptional activity of nuclear factor (NF)-κB, a major pro-inflammatory transcription factor, thereby suppressing pro-inflammatory gene expression (Figure 2) [26]. The activation of Epac1/2 also suppresses the synthesis of pro-inflammatory cytokines (Figure 2) [26].

Regarding specific anti-inflammatory targets of cAMP, annexin A1 (lipocortin-1) is an important one. Annexin A1 is an endogenous mediator of resolution of inflammation (pro-resolving mediator), mediating the anti-inflammatory effects of glucocorticoids such as phospholipase-A_2_ (PLA_2_) and arachidonic acid release inhibition, which prevents pro-inflammatory eicosanoid (prostaglandin, thromboxane, leukotriene) synthesis [26]. Annexin A1 is transcriptionally upregulated by cAMP via CREB (Figure 2) [31,32]. PKA also phosphorylates 5-lipoxygenase at Ser523 to induce synthesis of 15-epi-lipoxin A_4_, another potent pro-resolving mediator [33]. Interestingly, pro-resolving mediators can reciprocally increase cAMP levels, as well [26,34].

As far as specific pro-inflammatory mediators that are targeted by cAMP for inhibition are concerned, tumor necrosis factor (TNF)-α, IL-12, leukotriene B_4_ (LTB_4_), IL-1β, and various chemokines (CCL3, CXCL1, CCL2, CCL4, etc.) have been documented to be suppressed by cAMP-elevating agents (e.g., PDE4-selective inhibitors, such as apremilast, roflumilast, and crisaborole) [35,36,37,38,39,40,41,42,43,44]. In addition, PKA inhibits Ras homolog family member A (RhoA)-dependent integrin expression on the surface of granulocytes [45]. NF-κB inhibition by PKA (Figure 2) was one of the first anti-inflammatory actions of cAMP to be described more than 25 years ago [46,47,48]. Interestingly, PKA blocks NF-κB via different mechanisms depending on the cell type and inflammatory stimulus. For instance, PKA prevents IκBα proteasomal degradation, causing the cytosolic retention of NF-κB [49,50], or can prevent stimulatory signals from reaching NF-κB, such as phosphoinositide-3’-kinase (PI3K) and its effector kinase Akt (protein kinase B, PKB) [51]. It can also induce the formation of transcription-incompetent p50-p50 homodimers instead of the normal p50-p65 heterodimer that constitutes the transcriptionally active NF-κB complex [52,53].

The other major cAMP effector protein, Epac, plays a complementary but, nevertheless, equally important role in the suppression of inflammation as PKA does. Indeed, Epac also inhibits NF-κB [54] and induces the suppressor of cytokine signaling (SOCS)-3, which inhibits class I cytokine receptor signaling, such as the pro-inflammatory IL-6 receptor (Figure 2) [55,56,57]. The panel of substrates inhibited by Epac-induced SOCS3 via Rap1 includes the JAK/STAT1/3 pathway (Figure 2) [58]. Additionally, Epac activates the PI3K/Akt pathway, again via Rap1 (Figure 2), thereby inhibiting glycogen synthase kinase (GSK)-3β-mediated induction of the transcriptional repressor CCAAT displacement protein (CDP), which upregulates CCL3/4 levels [59].

Finally, it should be pointed out that, in addition to the particular cell type and pro-inflammatory stimulus studied, the anti-inflammatory signaling of cAMP also depends on its total amount present inside the cell. The overproduction of cAMP may lead to chemokine expression and release, rather than suppression, during monocyte differentiation to macrophages in vitro [60]. Thus, the balance between the activities of AC and PDEs needs to be tightly regulated in inflammatory conditions, as this balance ultimately determines the effects of the produced cAMP in the cell. The type of effector activated by cAMP, which can depend on the amount of cAMP present in the vicinity of the effector at any given time [61,62], is another important determinant. For instance, PKA inhibits PI3K/Akt in some cell types but Epac activates this pathway in others (see above). Thus, which effector’s activity prevails inside the cell may also determine the end-result of cAMP elevation. Given that Epac has a much lower sensitivity for cAMP than PKA does (despite similar affinities) [62], probably due to the positive cooperativity between the two cAMP-binding domains of the PKA regulatory subunits (Epac has only one nucleotide-binding domain), the amount of cAMP present may determine which effector is activated and, hence, the physiological outcome produced.

## 4. Cardiac GLP-1R-Dependent, cAMP-Mediated Cardio-Protective Signaling

A full discussion of the cardiovascular actions of the GLP-1R and its agonists (GLP1-RA’s) is beyond the scope of this review (see Refs. [6,13,14,16,18], for comprehensive reviews of this topic). In this section, we focus on the most important studies of cardiac-specific effects of the GLP-1R as they relate to cAMP-dependent signaling, with a particular attention to anti-inflammatory, cardio-protective actions. GLP-1R knockout (KO) mice have a reduced resting heart rate [63], supporting a role for GLP-1 in cardiovascular homeostasis regulation. Interestingly, in normal rats infused with exendin-4 in vivo, β-adrenoceptor (βAR)-dependent, but no αAR, activation of the sympathetic nervous system (SNS) was shown to be involved in GLP-1R’s cardiovascular effects of tachycardia and mild vasodilation [64]. The parasympathetic (cholinergic) branch of the autonomic nervous system may be involved as well [65]. GLP-1R KOs also suffer from diastolic dysfunction, increased left ventricular (LV) wall thickness, and impaired cardiac reserve compared to wild-type mice [64]. Given the central role of cAMP not only in the systolic (contraction) but also in the diastolic (relaxation) myocardial function [10], this cardiac phenotype in the absence of functional GLP-1R is not completely unexpected. Interestingly, studies on isolated perfused rat hearts and in cardiac myocytes suggest that GLP-1R-stimulated cAMP production may not be sufficient or capable of enhancing contractility, as βAR-stimulated cAMP normally does [66]; rather, GLP-1R-dependent cAMP elevation only enhances relaxation and diastolic function under basal conditions [66,67,68]. These studies suggest a negative inotropic action for GLP-1R-produced cAMP, which could be explained by differential, receptor-specific compartmentalization of cAMP signaling inside the cardiomyocyte, leading to opposite effects when stimulated by the GLP-1R or the β_1_AR [66]. This could also be explained by differential effector activation (PKA vs. Epac) by cAMP, depending on the stimulating receptor (GLP-1R or βARs, see also Section 3 above). However, a negative inotropic action for GLP-1R-produced cAMP is difficult to reconcile with the receptor’s well documented effects on heart rate (increased chronotropy, similar to βARs) in animals and in humans [6,14,16]. Additionally, alternative explanations for the phenotypic results of these studies are plausible, such as enhanced cholinergic stimulation of the M_2_ muscarinic receptor’s effects on the myocardium [65], cAMP-dependent stimulation of nitric oxide (NO) synthesis [67,68], a known negative inotrope in the heart [10], and, in studies using GLP1-RA with a catecholamine (isoprenaline) together, pharmacodynamic competition between βARs and GLP-1R for the same pool of cAMP-synthesizing ACs [66]. Even the increased heart rate induced by the atrial GLP-1R can indirectly reduce the contractile function of the ventricles [69]. Finally, it should be noted that, in the study by Vila Petroff et al. [66], which showed that GLP-1R-elicited cAMP failed to augment the contractility of adult rat cardiomyocytes, was ridden with a few experimental caveats that demand its results be treated with great caution: a relatively low GLP-1 concentration (only 10 nM) was used to stimulate cAMP synthesis, isoproterenol was not very efficient at stimulating cAMP (it caused only a doubling of basal cAMP levels, similar to GLP-1), and cAMP measurements appeared to be hindered by high PDE activity, as PDE inhibition with IBMX unmasked a very strong cAMP stimulation for βAR activation (from a 2-fold, with isoproterenol alone, to a 6-fold cAMP increase with isoproterenol plus IBMX) [66]. Notably, IBMX was unable to substantially increase GLP-1-stimulated cAMP production in this study [66], which, coupled with the fact that IBMX does not affect PDE8 (also cAMP-specific, like PDE4 and PDE7) [70], raises the intriguing possibility that GLP-1R-stimulated cAMP is tightly regulated by PDE8 in rat cardiomyocytes. In any case, it becomes clear from the aforementioned studies that the actual direct effect of cardiac GLP-1R-stimulated cAMP on contractility is far from being fully elucidated at present. 

In contrast, the role of GLP-1R-dependent cAMP signaling in protection against ischemic injury and apoptosis has been clearly demonstrated [6]. Via the activation of a variety of pro-survival kinases, such as PI3K/Akt, GSK-3β, ERK1/2, and p38 MAPK, GLP-1R-dependent cAMP leads to protective actions against cardiomyocyte apoptosis, oxidative stress, and inflammation (reviewed in Refs. [6,13]). However, the involvement of GLP-1R-independent signaling in some beneficial GLP1-RA effects on cardiac ischemia has also been suggested by some studies [6,13,14,71].

GLP-1R is expressed in the macrophages infiltrating the atherosclerotic plaque and can reduce lipid deposition and plaque volume via a 5’-adenosine monophosphate-activated protein kinase (AMPK)-independent action in streptozotocin-induced hyperglycemic and hyperlipidemic mice or type 1 diabetic rats [71]. Moreover, vascular smooth muscle GLP-1R activated by exendin-4 suppresses vascular smooth muscle cell proliferation and migration via the cAMP/PKA pathway in C57BL/6 mice [72]. Therefore, GLP-1R activation may stabilize atherosclerotic lesions via cAMP-dependent anti-inflammatory and anti-vascular remodeling mechanisms [73], which could position this receptor as an attractive anti-atherosclerotic target [74]. 

GLP-1R is expressed in human cardiomyocytes, more abundantly in atrial myocytes than in the ventricles [75]. There is also clinical evidence that GLP-1RAs protect against myocardial infarction (MI) occurrence [76], similar to how the beta-blockers do [77]. GLP-1R reduced MI size and inhibited cardiac-adverse remodeling (ventricular dilation, fibrosis, and hypertrophy) in a heart failure (HF) rat model in vivo [78]. However, these effects of GLP-1R activation by exendin-4 were reported to be mediated by the eNOS/NO/cGMP/PKG pathway and not by cAMP [78]. How exactly GLP-1R activated the eNOS/cGMP pathway was not investigated. Other cardio-protective signaling mechanisms against experimental MI elicited by the cardiac GLP-1R that have been reported thus far include estrogen-α receptor upregulation and IGF-1/IGF-2 modulation [79] and favorable extracellular matrix (pro-collagen Iα1/IIIα1, connective tissue growth factor, fibronectin, transforming growth factor (TGF)-β_3_) alterations and reduced inflammatory (IL-10, IL-1β, IL-6) gene expression, together with modulation of Akt/GSK-3β and Smad2/3 signaling [80,81]. In the latter study, exendin-4 also altered macrophage response gene expression in the absence of direct effects on cardiac fibroblasts, suggesting anti-inflammatory effects independently of anti-fibrotic effects [81].

In cardiomyocytes, GLP-1R activation inhibits TNF-α-induced apoptosis via cAMP/PKA and AMPK/SIRT1 pathways, which improve mitochondrial function and suppress pro-apoptotic Bax while upregulating anti-apoptotic Bcl-2 protein [82,83,84,85]. Additional anti-apoptotic mechanisms have been postulated for cardiac GLP-1R, including Notch signaling activation [86] and oxidative stress relief via the mTORC1/p70 ribosomal protein S6 kinase (p70S6K) pathway [87] but also others [88]. Unfortunately, the involvement (or not) of cAMP in any of these pathways was not examined but is certainly possible. GLP-1R also modulates intracellular calcium homeostasis to protect against ischemia/reperfusion injury in cardiomyocytes [89]. There is also evidence for a potential anti-arrhythmic effect of the cardiac GLP-1R via enhanced sarcoplasmic reticulum calcium re-uptake and reduced calcium release via ryanodine receptors type 2 (RyR2) in ventricular arrhythmia, as well as suppression of the pro-arrhythmic CaMKII activity via cAMP (Figure 2) [90]. It could also activate ATP-sensitive K^+^ channels (hyperpolarizing K_ATP_ channels), which are known to be activated by PKA [91], to attenuate pressure overload-induced cardiac apoptosis and hypertrophy and arrythmias [65,92]. Nevertheless, GLP-1RAs have also been associated with tachycardia in HF patients [93,94], so their net effect on heart rate is rather complicated and can be unpredictable. 

Hyperglycemia-mediated cardiomyocyte damage has a significant inflammatory component and GLP-1R can protect against inflammation and endoplasmic reticulum (ER) stress via NFκB inhibition in diabetic hearts [95]. GLP-1R has also been reported to protect against cardiac hypertrophy promoted by angiotensin II (AngII) through inhibition of the nicotinamide adenine dinucleotide phosphate-oxidase (NOX)-4/histone deacetylase (HDAC)-4 pathway [96]. Interestingly, and despite the fact that its presence in cardiac fibroblasts has yet to be verified experimentally, GLP-1R has been reported to inhibit the activation (fibrosis) of this tissue type, as well. Specifically, GLP1-RAs may inhibit the Janus kinase (JNK)-Activator protein (AP)-1 signaling pathway via prolyl 4-hydroxylase downregulation and reactive oxygen species (ROS) production induced by the pro-fibrotic hormone AngII, thereby reducing cardiac fibroblast proliferation/activation and cardiac fibrosis [97,98]. Moreover, liraglutide was reported to inhibit AngII- and glucose-induced collagen synthesis via NFκB and ERK1/2 suppression in cardiac fibroblasts [99]. Finally, liraglutide was shown to ameliorate cardiac fibrosis secondary to abdominal aortic constriction in vivo through AngII type 1 receptor (AT_1_R) (the pro-fibrotic one) downregulation and AT_2_R (the cardioprotective AngII receptor type) upregulation [100]. Unfortunately, the role of cAMP signaling in any of these effects of GLP-1R agonism was investigated in none of these studies, so the extent to which these anti-oxidant and anti-fibrotic effects of GLP-1R were cAMP-dependent remains undetermined. However, since cAMP is known to mitigate cardiac fibrosis via both PKA- and Epac-mediated mechanisms [101,102,103,104,105], it is rather plausible that cAMP plays an essential role in the anti-fibrotic actions of the GLP-1R.

Pyroptosis is an important cellular process linking inflammation with apoptosis and rather often underlies diabetes-associated cardiomyocyte loss. Exendin-4 was reported to protect against cardiac remodeling and inflammation in high-fat diet (HFD)-fed rats by promoting AMPK phosphorylation, which, in turn, downregulated thioredoxin-interacting protein (TXNIP, an oxidative stress-related protein), pro-apoptotic caspase-1, pro-inflammatory IL-1*β* and IL-18, and, finally, pyroptosis in primary murine cardiomyocytes exposed to high glucose in vitro [104]. More evidence of a protective effect against cardiac pyroptosis was added by another study showing that GLP-1R activation with liraglutide inhibited TNFα- and hypoxia-induced NOD-like receptor family pyrin domain-containing (NLRP)-3 inflammasome activation in the rat cardiomyoblast cell line H9c2 [105]. Mechanistically, liraglutide activated SIRT1, lowering NOX4 activity, ROS levels, and NLRP3 inflammasome expression in TNFα- and hypoxia-treated H9c2 cardiomyocytes [105]. Again, the role of cAMP in this effect of liraglutide was not directly examined but it should be noted that PKA phosphorylates and inhibits NLRP3 inflammasome activation [106,107].

Similarly to liraglutide, semaglutide also activated the AMPK pathway, improving autophagy, oxidative stress, and survival of H9c2 cardiomyocytes treated with the major proinflammatory stimulus LPS (the lipopolysaccharide toxin) [108]. All myocardial injury markers and morphological changes, including lipid accumulation area, were ameliorated and NF-κB, TNF-α, and IL-1β levels were lowered by semaglutide in LPS-exposed H9c2 cells [108]. The same was true for dulaglutide, which also ameliorated LPS-induced oxidative stress by suppressing mitochondrial ROS levels and downregulating NOX1 in H9c2 cells [109]. Interestingly, dulaglutide lowered creatine kinase isoenzyme-MB (CK-MB) and cardiac troponin I (cTnI) and, importantly, also suppressed iNOS expression, NO production, and TNF-α, IL-1β, IL-17, matrix metalloproteinase (MMP)-2, and MMP-9 expression, all important mediators of LPS-induced inflammation. All these effects were mediated by GLP-1R-induced NFκB inhibition [108]. Unfortunately, neither study examined the cAMP involvement in the effects of semaglutide or dulaglutide again. In contrast, a recent study on the protective effects of exendin-4 in H9c2 cells exposed to the mitochondrial disruptor methylglyoxal uncovered a direct role for cAMP in the protection against methylglyoxal-induced mitochondrial dysfunction, apoptosis, oxidative stress, and inflammation afforded by the GLP-1R via its effector Epac [110]. Specifically, the authors of this study reported that exendin-4-induced cAMP directly activates the anti-apoptotic and mito-protective PI3K/Akt pathway in H9c2 cardiomyocytes by activating Epac, which, in turn, activates Rap1, a small GTPase known to stimulate PI3K/Akt in various cell types (Figure 2) [111]. Thus, GLP-1R can suppress cardiac apoptosis and inflammation not only via cAMP-activated PKA but also through cAMP-activated Epac1/2 (Figure 2).

## 5. Sympathoadrenal GLP-1R-Dependent, cAMP-Mediated Signaling

There is a close interplay between the GLP-1R and the SNS in the CNS. GLP-1R is involved in hypothalamic–pituitary–adrenal (HPA) axis activation. Indeed, exendin-4, which actively elevates circulating ACTH and corticosterone levels in rats, potently activates the HPA axis [112]. Conditional GLP-1R knockdown, specifically in paraventricular nucleus (PVN) neurons in the CNS, reduces the HPA axis response to acute and chronic stress [113]. The SNS innervates the adrenal cortex increasing ACTH-dependent steroidogenesis and GLP-1R activation enhances this effect [114], while also stimulating the adrenal medulla to increase catecholamine secretion [115]. Central GLP-1R activation increases SNS outflow to the heart to elevate blood pressure and heart rate, an effect abolished by βAR antagonists (beta-blockers) like propranolol or atenolol [116]. Direct SNS activation by the central GLP-1R has been suggested for GLP1-RAs’ positive chronotropy [117], as well as the attenuation of parasympathetic neurotransmission to the heart [118]. In humans, the net effect of GLP-1R agonism on heart rate is more complicated, with some studies claiming an increase in healthy volunteers and obese or diabetic patients but showing no effect in others [119,120].

Although cAMP exerts powerful anti-inflammatory effects on immune cells and GLP-1RAs are known to inhibit systemic inflammation, GLP-1R is not widely expressed in the immune system [121], which suggests that GLP-1R-stimulated cAMP inside immune cells is unlikely to directly impact inflammation substantially. A very recent study by Drucker and colleagues provided important insights into the actions of GLP-1R agonism on systemic inflammation [15]. These authors showed that central neuronal GLP-1R activation is essential for the reduction in circulating TNFα levels induced by multiple Toll-like receptor agonists (major pro-inflammatory stimuli) [15]. Importantly, the authors also uncovered that this anti-inflammatory effect of GLP-1R activation in the CNS required α_1_ARs, δ- and κ-opioid receptors [15]. This study had some major limitations, such as the use of plasma TNFα levels only as marker of systemic inflammation and significant variations in the exendin-4-induced reduction in these levels among the controls of its various experimental sets; for example, α_1_AR antagonism with prazosin resulted in marked TNFα reduction on its own, minimizing any additional reduction by exendin-4. On the other hand, bilateral adrenalectomy and β_2_AR antagonism with ICI-118,551 alone caused large TNFα increases, which masked the reduced responses to exendin-4. In other words, adrenalectomy (i.e., cortisol and adrenaline) and the β_2_AR probably still played significant roles in the anti-inflammatory effects of the central neuronal GLP-1R. Nevertheless, the findings of this study are very important since they represent the first demonstration of the impact the brain GLP-1R has on systemic inflammation via mechanisms unrelated to immune cells per se. 

The GLP-1R was recently shown to be present and functional also in adrenal chromaffin cells, where it modulates catecholamine (epinephrine and norepinephrine) secretion via the regulation of exocytosis (Figure 3) [122]. Adrenal catecholamines are essential in the body’s response to stress [123] and, particularly, the hormone adrenaline (epinephrine) they uniquely (sympathetic neurons only release norepinephrine, the SNS neurotransmitter) secrete is vital for the generation of the classic “fight or flight” response of the body in situations of acute stress or danger [124]. Catecholamine secretion from the chromaffin cells is an example of the classic exocytotic process involving vesicle fusion with the cell membrane in a calcium-dependent manner [124,125,126,127,128]. Machado and colleagues reported that GLP-1R is expressed in bovine chromaffin cells and, interestingly enough, was found accumulating in the cytoplasm to a significant extent, which may suggest increased turnover of the receptor in response to the homeostatic status of the adrenal medulla (e.g., presence of stress signals, etc.) [122]. Moreover, the authors found that, although it is not a direct secretagogue of catecholamines, chromaffin cell GLP-1R enhances secretion in response to cholinergic stimulation of nicotinic cholinergic receptors (the physiological stimulus of adrenal catecholamine secretion), as well as the synthesis of catecholamines via tyrosine hydroxylase upregulation [122], the enzyme catalyzing the rate-limiting step in catecholamine biosynthesis [124] (Figure 3). GLP-1R did not appear to increase the number of secretory vesicles but rather increased the amount of catecholamine released by each individual vesicle (the so-called “quantum size”) [122]. Finally, the authors found that this positive effect of GLP-1R on catecholamine secretion is mediated by cAMP-activated PKA and not by Epac1/2 (Figure 3), and thus, differs from GLP-1R’s positive modulation of insulin secretion in pancreatic β-cells, which is mediated by both PKA and Epac [118]. This effect of GLP-1R in the adrenal medulla is consistent with reports of strong stress response potentiation (an “amphetamine”-like effect) by this receptor [129]. At the same time, however, it seems paradoxical given the opposite effects of GLP-1 and catecholamines on the regulation of blood glucose levels (catecholamines are pro-hyperglycemic). Nevertheless, pancreatic β-cells express adrenergic receptors of the β_2_ subtype (β_2_ARs), which enhance insulin secretion [130], contrary to their α_2_AR counterparts that inhibit it. Adrenaline, which is only secreted by the adrenal medulla, has a particularly high affinity (much higher than that of the SNS neurotransmitter norepinephrine) for the β_2_AR subtype [131]. Therefore, if it occurs also in vivo, the adrenal GLP-1R-dependent facilitation of epinephrine secretion may serve to fine-tune pancreatic insulin release, especially in situations of acute or extreme stress.

## 6. Conclusions/Future Perspectives

Thanks in large part to their growing popularity in pop culture and in social media, and to the aggressive marketing of GLP1-RA medication manufacturers, the field of GLP-1R pharmacology/biology has seen an explosion over the past 6 years or so. Although GLP-1R pharmacology appears simple enough, i.e., only one receptor for GLP-1 and its analogs, signaling only via one modality (Gs/AC/cAMP), there is still a lot to be learned and delineated about this receptor and its ligands.

The biggest questions awaiting answers in future studies are as follows: (a) Which actions of the GLP-1R are mediated by the receptor in the CNS and which by peripheral tissue-expressing receptor; (b) Whether all GLP-1R agonists share all the GLP-1R actions reported thus far or not; (c) What other receptors can be targeted simultaneously with the GLP-1R by the same agonist (current multi-agonists target two or three different receptors, e.g., tirzepatide, retatutride [21,25], but could ligands that target more than three be on the horizon?); and (d) What type(s) of medications can be combined with GLP1-RAs for an additive or synergistic effect and for which disease/condition. For example, in the case of anti-inflammatory treatment, the role of cAMP appears, from all the studies discussed above, to be rather prominent; thus, perhaps cAMP-elevating medications, such as the PDE4 inhibitors already used in inflammatory conditions, like atopic dermatitis and chronic obstructive pulmonary disease (COPD), could be combined with a GLP1-RA for an enhanced anti-inflammatory effect, especially in diabetic or obese patients. Indeed, emerging data from our lab point to an enhanced anti-inflammatory effect of the GLP-1R by PDE4 inhibition in cardiac myocytes (manuscript under review). Such a combination therapy would be particularly important in the context of diseases characterized by suppressed cAMP production in the diseased tissue, such as is the case for the chronically failing human heart (10) in the context of chronic human HF. Alternatively, the possibility that GLP-1R uses cAMP-independent signaling mechanisms in some cell/tissue types [68,78], if it holds true for humans in vivo, may give the GLP1-RAs a therapeutic edge over other pharmacotherapies for obesity- or diabetes-associated human HF (cardiomyopathy), an indication they recently started being used for in clinical practice.

One more vital question regarding specifically the GLP1-RAs’ cardiovascular benefits is to what extent the cardiovascular actions of GLP-1R are direct, i.e., emanating from cAMP signaling elicited by GLP-1Rs residing in cardiovascular tissues, or indirect, i.e., secondary to GLP-1R-dependent cAMP signaling’s anti-diabetic, metabolic, and anti-obesity actions. Obviously, this will depend on the particular effect in question and most (if not all) effects will probably be both direct and indirect. In any case, the possibilities for expanding the therapeutic indications of the GLP1-RA agents appear to be enormous, making the future surrounding GLP-1R pharmacology truly exciting. As astonishing as it may sound, the current surge in studies pertaining to this receptor is only the beginning.

## Figures and Tables

**Figure 1 pharmaceutics-16-00693-f001:**
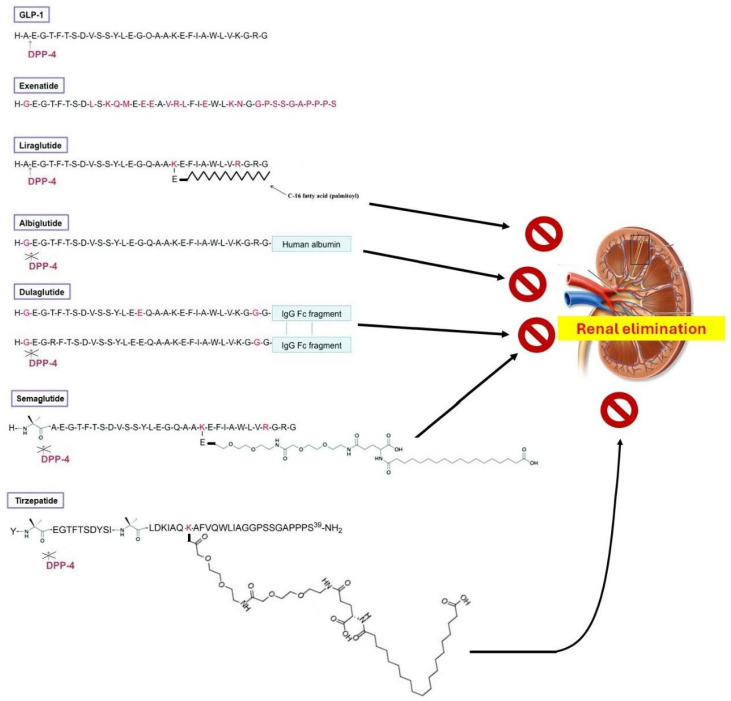
Examples of GLP1-RAs with their most important structural features. See text for details.

**Figure 2 pharmaceutics-16-00693-f002:**
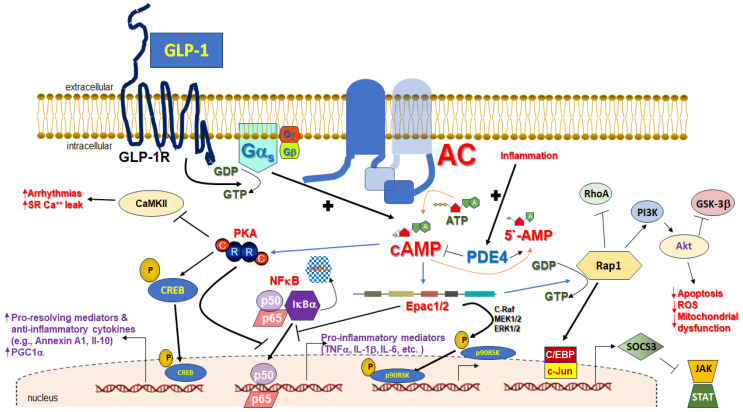
Cardiac GLP-1R cAMP-mediated signaling toward reduced inflammation. For simplicity, only the cAMP-mediated signaling pathways elicited by the GLP-1R in cardiac cells and their effectors related to inflammation modulation are shown. PKA is a tetramer consisting of two regulatory (R) and two catalytic (C) subunits. The active form of NFκB consists of two subunits p50 and p65, which, at rest, are retained in the cytoplasm by the IκBα subunit. Upon an appropriate stimulus (pro-inflammatory), IκBα becomes phosphorylated and ubiquitinated, i.e., targeted for proteasomal degradation, which frees the p50/p65 dimer to translocate to the nucleus and activate gene transcription. See text for more details. Acronyms not explained in the main text: 5’-AMP: 5’-adenosine monophosphate; ATP: Adenosine triphosphate; CaMKII: Calcium/calmodulin-dependent protein kinase II; C/EBP: CCAAT-enhancer-binding protein; c-Raf: cellular (protooncogene) Rapidly accelerated fibrosarcoma; c-Jun: c-Jun transcription factor; ERK1/2: Extracellular signal-regulated kinase-1/2; GSK-3β: Glycogen synthase kinase -3beta; GDP: Guanosine diphosphate; GTP: Guanosine triphosphate; Gα_s_: Stimulatory alpha subunit of heterotrimeric guanine nucleotide-binding protein; JAK: Janus kinase; MEK1/2: Mitogen-activated protein kinase kinase-1/2; PKA-C: Catalytic subunit of PKA; PKA-R: Regulatory subunit of PKA; PGC1α: Peroxisome proliferator-activated receptor gamma co-activator 1alpha; RhoA: Ras homolog family member A; ROS: Reactive oxygen species; p90RSK: p90 ribosomal S6 kinase; SR: Sarcoplasmic reticulum; STAT: Signal transducer and activator of transcription; SOCS3: Suppressor of cytokine signaling-3. “P” enclosed in a dark yellow circle indicates phosphorylation, arrows indicate stimulation, upward arrows or “+” indicate increase, “—I” indicate inhibition.

**Figure 3 pharmaceutics-16-00693-f003:**
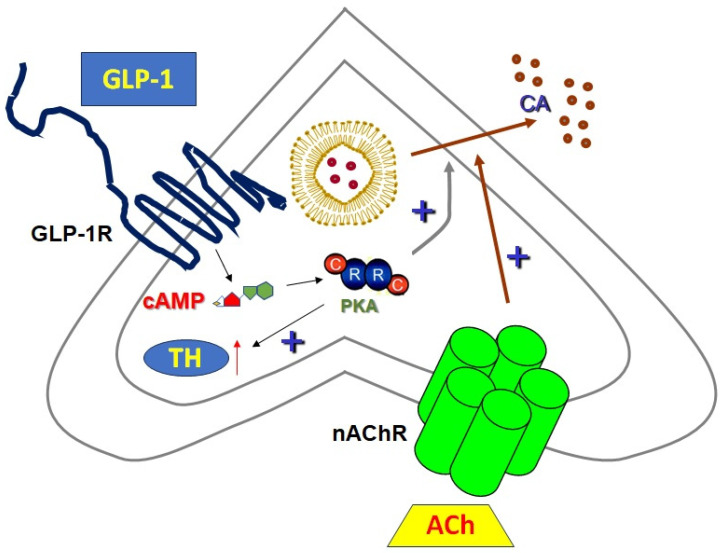
Adrenal chromaffin cell GLP-1R cAMP-mediated signaling toward catecholamine secretion. nAChR: Nicotinic cholinergic receptor; ACh: Acetylcholine; CA: Catecholamine (norepinephrine or epinephrine); TH: Tyrosine hydroxylase. Arrows with “+” next to them indicate stimulation/potentiation.

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
