# Peer review of "Cyclic Adenosine Monophosphate in Cardiac and Sympathoadrenal GLP-1 Receptor Signaling: Focus on Anti-Inflammatory Effects"

_pharmaceutics, 2024, doi:10.3390/pharmaceutics16060693_

Round 1

Reviewer 1 Report

Comments and Suggestions for Authors

The manuscript entitled "Cyclic adenosine monophosphate in cardiac and sympathoadrenal -GLP-1 receptor signaling: Focus on anti-inflammatory effects" by Lymperopoulos et al reviews the current knowledge on the involvement of cAMP in the GLP-1R signaling. The topic of this subject is contemporary, several reviews on this topic appearing frequently, showing an ongoing interest and research on characterizing new GLP-1R involvement in pathology and the discovery of new pharmaceutical targets along this signaling pathway.

The manuscript was carefully documented and written.

Comments:

·        Some information may have been reviewed in previous publications, however, there is more important to cite original papers too. One example is at L83, where references 13 and 14 are both reviews.

·        L103-104: Heloderma suspectum should be italicized.

·        L145 – a reference is missing regarding the structure of GLP-1RA.

·        Figure 1 – I suggest a bigger font for the text on the figure.

·        L224 – correct “THE” to “The”

·        Chapter 3: “cAMP in inflammation” – requires a comment about the sensitivity of PKA vs EPAC towards cAMP, to complete the discussion about the differences in cell responses depending on cAMP levels, as emphasized at L229-230.

·        Legend of Figure 2 is very detailed in comparison to the legend of Figure 1 or 3.  I suggest a legend emphasizing the link between cAMP and inflammation describing the PKA and EPAC effects on NFkB activation, and not the general mechanisms of PKA and NFkB activation. Moreover, the diagram shows a direct effect of PKA on translocated p50 subunit, which is not explained in the text at the corresponding paragraph (L211-219). Reference 45 describes a PKA effect on p65, while the figure suggests an effect on p50 subunit.

·        The first paragraph of Chapter 4 may need some revision to present better the information regarding the competition of GLP-1R with βAR receptors for cAMP that results either in an additive or inhibitory effect. In the text from L266 to L269 the two cited studies 60 and 61 were conducted on different animal models: GLP-1R KO mice and rats with normal levels of GLP-1R, respectively. L274 – “this cardiac phenotype… is completely expected” – is a statement that suggest that the cAMP produced after GLP-1R activation is more important than cAMP produced via β-adrenergic receptors for cardiac function, and contradicts the next sentence.  Please revise this paragraph so that the message is more clear. L276 – typo “stumulated”

·        Would authors like to comment whether GLP-1R antagonists have been also tested in vivo or in vitro in relation to cAMP levels or the degree of pro- or anti-inflammatory effects?

Author Response

Some information may have been reviewed in previous publications, however, there is more important to cite original papers too. One example is at L83, where references 13 and 14 are both reviews.

Author response: We thank this reviewer for the overall kind and positive comments about the quality of our work. This is a pertinent remark. We have cited a couple of reviews here because this part of the text belongs to the “Introduction”, wherein we merely introduce the various beneficial actions of GLP1-RAs. Their anti-inflammatory actions are discussed in detail later in the text with the appropriate original research articles cited alongside.

  • L103-104: Heloderma suspectum should be italicized.

     Author response: Done.

  • L145 – a reference is missing regarding the structure of GLP-1RA.

     Author response: Reference added, thank you.

  • Figure 1 – I suggest a bigger font for the text on the figure.

     Author response: Done, thank you.

  • L224 – correct “THE” to “The”

     Author response: Done, thank you.

  • Chapter 3: “cAMP in inflammation” – requires a comment about the sensitivity of PKA vs EPAC towards cAMP, to complete the discussion about the differences in cell responses depending on cAMP levels, as emphasized at L229-230.

     Author response: This is an excellent point, and we thank this reviewer for raising it. Epac and PKA have indeed been reported to exhibit differential sensitivities to cAMP (new Refs. 61 & 62 for the revised manuscript), which may explain, in part, their differential (even opposing) actions in cells in several physiological/pathological situations. Accordingly, we have now added a few sentences in the revised text (lines 244-246 & 248-253 of the revised manuscript, highlighted in yellow) discussing this very interesting point (and we also cite these two pertinent papers: new Refs. 61 & 62 of the revised manuscript).

  • Legend of Figure 2 is very detailed in comparison to the legend of Figure 1 or 3.  I suggest a legend emphasizing the link between cAMP and inflammation describing the PKA and EPAC effects on NFkB activation, and not the general mechanisms of PKA and NFkB activation. Moreover, the diagram shows a direct effect of PKA on translocated p50 subunit, which is not explained in the text at the corresponding paragraph (L211-219). Reference 45 describes a PKA effect on p65, while the figure suggests an effect on p50 subunit.

     Author response: Thank you for the suggestion; we just added a brief sentence in Fig. 2 legend to explain how PKA is activated by cAMP, since this is not mentioned in the text. However, the point of this reviewer that this might be unnecessary details is well taken and thus, we have now removed those details from the legend of that figure in the revised manuscript. Regarding a “direct effect of PKA on p50” shown in the figure, that was not our intention and we apologize for the confusion. The inhibition symbol was meant to simply indicate inhibition of NFκB nuclear translocation by PKA. We have now tweaked the schematic illustration to make this clearer in the revised manuscript. We hope this satisfies now this reviewer.

  • The first paragraph of Chapter 4 may need some revision to present better the information regarding the competition of GLP-1R with βAR receptors for cAMP that results either in an additive or inhibitory effect. In the text from L266 to L269 the two cited studies 60 and 61 were conducted on different animal models: GLP-1R KO mice and rats with normal levels of GLP-1R, respectively. L274 – “this cardiac phenotype… is completely expected” – is a statement that suggest that the cAMP produced after GLP-1R activation is more important than cAMP produced via β-adrenergic receptors for cardiac function, and contradicts the next sentence.  Please revise this paragraph so that the message is more clear. L276 – typo “stumulated”

      Author response: We thank this reviewer for another excellent suggestion and apologize for the ambiguity of this paragraph in the original manuscript text. We have now completely revised this paragraph to correct these errors in the original text and to get our message across more clearly (lines 281-324 of revised text, also highlighted in yellow). We hope this paragraph is clearer now to this reviewer.

  •  

         Would authors like to comment whether GLP-1R antagonists have been also tested in vivo or in vitro in relation to cAMP levels or the degree of pro- or anti-inflammatory effects?

     Author response: Thank you for this pertinent remark. However, we are unaware of any studies on GLP-1R antagonists in relation to cAMP production or effects in inflammation.

Reviewer 2 Report

Comments and Suggestions for Authors

This manuscript reviews literature that studied the GLP-1 receptor signaling, mainly in the cardiac tissue and the sympathoadrenal system. This also includes a review of current synthetic GLP-1R agonists and signal transduction cascades downstream of GLP-1R. The manuscript is a list of published facts, not much of a deeper discussion. However, this manuscript is backed by a large number of references. This paper presents good reading material for entry-level scientists in the field. There are a few texts that are difficult to understand or include some typos. The following are specific points to be considered.

L221-223. Fig. 2 does not include the IL-6 receptor. Either move (Figure 2) behind (SOCS)-3 or add more text.

L230-232. This is misleading since Ref 59 specifically discusses the differentiation period from monocytes to macrophages. This should be included here.

L251. Need “;” after fibrosarcoma.

L313. The abbreviation HF should be defined.

L335. “reduced ryanodine receptor type 2 (RyR2) leak”. This should be “reduced release through ryanodine receptor type 2 (RyR2)”.

 L347-349. Need “that” after "fact." It is not clear what “favorably modulate” means. Please revise this.

L489. “current multi-agonists target three receptors.” Please name these agonists and receptors with appropriate references.

Comments on the Quality of English Language

Some difficult text. there are a few very long sentences that can be split.

Author Response

This manuscript reviews literature that studied the GLP-1 receptor signaling, mainly in the cardiac tissue and the sympathoadrenal system. This also includes a review of current synthetic GLP-1R agonists and signal transduction cascades downstream of GLP-1R. The manuscript is a list of published facts, not much of a deeper discussion. However, this manuscript is backed by a large number of references. This paper presents good reading material for entry-level scientists in the field. There are a few texts that are difficult to understand or include some typos. The following are specific points to be considered.

L221-223. Fig. 2 does not include the IL-6 receptor. Either move (Figure 2) behind (SOCS)-3 or add more text.

Author response: We thank this reviewer for the overall kind and positive comments about the quality of our work. However, we are not sure we understand this comment. Why does the IL-6 receptor need to be included in Fig. 2? In any case, IL-6 is included in the figure as a downstream target gene of NFκB and, as discussed in the text, Epac is shown to inhibit NFκB and to activate SOCS3 via Rap1 in Fig. 2.

L230-232. This is misleading since Ref 59 specifically discusses the differentiation period from monocytes to macrophages. This should be included here.

Author response: Done, thank you.

L251. Need “;” after fibrosarcoma.

Author response: Done, thank you.

L313. The abbreviation HF should be defined.

Author response: Done, thank you.

L335. “reduced ryanodine receptor type 2 (RyR2) leak”. This should be “reduced release through ryanodine receptor type 2 (RyR2)”.

Author response: Done, thank you.

 L347-349. Need “that” after "fact." It is not clear what “favorably modulate” means. Please revise this.

Author response: Done, thank you.

L489. “current multi-agonists target three receptors.” Please name these agonists and receptors with appropriate references.

Author response: Done, thank you. All changes to the revised text, according to these comments, are highlighted in yellow.